# High-Risk Sarcoidosis: A Focus on Pulmonary, Cardiac, Hepatic and Renal Advanced Diseases, as Well as on Calcium Metabolism Abnormalities

**DOI:** 10.3390/diagnostics14040395

**Published:** 2024-02-11

**Authors:** Dominique Israël-Biet, Nicol Bernardinello, Jean Pastré, Claudio Tana, Paolo Spagnolo

**Affiliations:** 1Service de Pneumologie et Soins Intensifs, Hôpital Européen Georges Pompidou, Assistance Publique-Hôpitaux de Paris, 75015 Paris, France; 2Department of Cardiac, Thoracic, Vascular Sciences and Public Health, University of Padova, 35128 Padova, Italy; nicol.bernardinello@unipd.it; 3Geriatrics Clinic, SS Annunziata University-Hospital of Chieti, 66100 Chieti, Italy; 4Section of Respiratory Diseases, University of Padova, 35121 Padova, Italy

**Keywords:** sarcoidosis, high-risk sarcoidosis, sarcoidosis mortality, sarcoidosis morbidity

## Abstract

Although sarcoidosis is generally regarded as a benign condition, approximately 20–30% of patients will develop a chronic and progressive disease. Advanced pulmonary fibrotic sarcoidosis and cardiac involvement are the main contributors to sarcoidosis morbidity and mortality, with failure of the liver and/or kidneys representing additional life-threatening situations. In this review, we discuss diagnosis and treatment of each of these complications and highlight how the integration of clinical, pathological and radiological features may help predict the development of such high-risk situations in sarcoid patients.

## 1. Introduction

Sarcoidosis is generally viewed as a benign systemic disease, although differently affecting subjects according to their race, sex, age and socioeconomic status, with a worse overall prognosis when the latter is low and in Black patients [1,2]. In several clinical settings, the disease will progress no matter the treatment, and will directly impact the prognosis through what is now known as high-risk sarcoidosis. This comprises either a specific unique vital organ or a multiorgan involvement or even a dangerous metabolic situation (hypercalcemia, for instance). The literature is continuously enriched with new data on sarcoidosis pathogeny, even though its origin is still largely unknown, ultimately contributing to an optimized clinical management. Recent updated recommendations have been published on the diagnosis and treatment of sarcoidosis [3,4]. High-risk sarcoidosis settings are also increasingly recognized [5,6,7,8]. We focused in this review on several contributors to sarcoidosis mortality and morbidity: advanced pulmonary, cardiac, hepatic and renal sarcoidosis, as well as calcium metabolism abnormalities, in order to highlight the main comprehensive and updated steps to their management.

## 2. Advanced Pulmonary Sarcoidosis (APS)

This term is used to refer to sarcoidosis phenotypes which lead to significant risks of loss of lung function, respiratory failure or death. It is mainly due to the development of severe pulmonary fibrosis and to its complications, such as pulmonary hypertension, bronchiectasis, infections and acute exacerbations [5,6,7,8].

A significant factor of mortality [5,9,10,11,12,13,14,15], its prevalence is 10–20% [2,10,16]. Among the general sarcoidosis population, the great majority will have a complete remission, while about one third may progress to chronic and progressive disease. Most sarcoid patients with a low degree of pulmonary fibrosis will remain relatively stable. The progression to an extensive fibrosis is at least partly associated with environmental/occupational factors [17,18,19,20] and gene–environment interactions [18,21,22,23]. Its treatment is not well delineated and relies upon anti-inflammatory drugs if appropriate, antifibrotic ones in particular cases, and supportive care in any case. The role of vasodilators in SAPH is still under investigation. Lung transplantation might be indicated in end-stage disease, and will be discussed with experts in order to define the most appropriate period to refer these patients to specialists.

### 2.1. Clinical Features

APS diagnosis is typically reached at the age of 48 years [12], either at first presentation [24,25] or after a variable follow-up, usually >5 years [26]. None of the clinical sign or symptom is specific of the condition, with the most common ones being dyspnea (80%), cough (51%) and, rarely, hemoptysis (3%). Crackles can be present in patients with traction bronchiectasis (28%) and wheezing (6%) in those with airway obstruction [12]. Nail clubbing is very rare (<10% of patients) [27].

### 2.2. Pulmonary Function Tests

APS can present with varying PFT features, such as a restrictive, obstructive or mixed patterns, associated with gas-exchange defects [8,28,29,30]. In a study of a very large cohort, airflow obstruction was noted in 26% and mixed ventilatory defect in 10% of the sample. The latter defect was associated with a decreased DLCO, a stage 4 disease and a higher mortality [29]. The extent of fibrosis on HRCT is correlated to the degree of ventilatory restriction [29]. Airway obstruction can be found in any disease stage, but is more frequent in stages 3 (10–23%) and 4 (45–71%) associated with airway traction and stenosis [29,30]. DLCO, frequently reduced in APS whatever its functional pattern (restrictive, obstructive or mixed), is the only independent functional predictor of mortality. The abnormalities of TM6 are correlated with the decrease in DLCO [31,32], and reflect the patients’ functional impairment, decreased quality of life and clinical symptoms. The 6 mn walk distance (6MWD) is decreased in these patients, correlating of a decreased FVC and oxygen saturation on exercise [31]. It should be noted here that both reduced DLCO and 6MWD are strongly predictive of pulmonary hypertension.

### 2.3. Imaging

Chest X-ray: the most common features are an association of hilar lymphadenopathy and various parenchymal changes, such as reticular or reticulonodular opacities projecting from the hilum [24,33], typical of Scadding stage 2, or without lymphadenopathy (stage 3), predominantly in the upper zones, with more or less important traction bronchiectasis (fibrotic lesions, stage 4).

High resolution computed tomography (HRCT) more precisely reflects the changes underlying the PFT defects and three major patterns of APS can be observed, variously associated [34,35,36]: central bronchial distortion, observed in roughly half of the patients; diffuse hilar linear opacities, observed in 24% of cases [36], usually irreversible but causing a slight functional impairment; and peripheral upper zones honeycombing [36,37], usually associated with decreased lung volumes and DLCO and increased oxygen requirements. Fibrocystic opacities, larger than honeycombing, are found along the airways to the peribronchovascular regions, as well as to the fissures [5,38]. They can be associated with paracicatricial emphysema [37,38], sometimes sheltering a mycetoma. HRCT can also show small nodules in the peribronchovascular interstitium [39], larger nodules, ground-glass or consolidation opacities predominantly in the mid- and upper zones. More characteristic of the APS fibrotic changes: airway distortion with traction bronchiectasis and airway angulation. Finally, ongoing inflammatory processes within bronchial walls can cause their thickening as well as reduced lumens. The Right Upper Lobe Bronchus Angle (RUL-BA) has been reported to be larger in patients with a stage 4 sarcoidosis than in those without [40]. This measure might assist radiologists in diagnosing a fibrotic sarcoidosis. Studies are ongoing to evaluate the predictive value of this measurement on progression. Among a very large cohort of sarcoid patients, those who progressed to chronic respiratory failure were retrospectively analyzed for CT findings of pulmonary fibrosis [41,42]. As the latter progressed, the combination of the above described features led to a shrinkage of upper lobes. Interestingly, patients with multiple peripheral cysts/bullae had a unique disease course characterized by wheezing and concomitant pulmonary hypertension and mycetoma [42]. Finally, CT findings can, per se, have a significant prognostic value. Indeed, patients with a lower lung-dominant sarcoidosis are more at risk of progression, acute deterioration and higher long-term mortality than those with classical upper lobes lesions [43]. Those with an aspect of combined sarcoidosis and idiopathic pulmonary fibrosis, whether or not this is a distinct new phenotype or an evolutive aspect of APS in subjects with a particular genetic background, have a darker prognosis and even though under antifibrotic drugs progress more rapidly than IPF patients [44,45]. Finally, a recent and comprehensive study of patients with stage 4 pulmonary sarcoidosis showed that an extension of fibrotic lesions on HRCT > 20%, upper lobes fibrocystic changes and basal subpleural honeycombing were associated with a worse function and a worse survival, with honeycombing being the sole imaging feature to be an independent predictor of reduced survival [46].

### 2.4. F-FDG PET/CT

This is very sensitive in identifying active inflammation in sarcoidosis, and can be key to the delineation of therapeutic strategies in APS patients [6,47] by allowing us to distinguish between end-stage, inactive fibrosis and that in which anti-inflammatory drugs might have some benefit [48,49]. One study has shown that the majority of APS patients with all features of pulmonary fibrosis on HRCT had PET positive findings, even in those without any sign of serologic inflammation [50]. However, no SUV threshold has proved so far to be reliable in diagnosing active inflammation inside fibrotic lesions [5], and disease activity evaluated by HRCT and ^18^F-PET/CT is discordant most of the time [51,52].

### 2.5. Innovative Imaging Modalities

This is not routinely used nor precisely evaluated in sarcoidosis, and some newer techniques might have future interesting developments as diagnosis and prognostic tools in APS [53]. Magnetic resonance, for instance, can differentiate areas of active inflammation from fibrosis [54], while radiomic HRCT measures differentiate disease abnormalities, identify trends in Scadding stages and are associated with lung function characteristics [55]. They might strongly help clinicians in the future in better phenotyping APS subsets and monitor the disease course.

### 2.6. Biomarkers

None of them presently provides clinicians with a significant value in the management of APS, either due to their low sensitivity/specificity or because they are not routinely available in clinical facilities. C-reactive protein (CRP), a non-specific marker of inflammation, is particularly increased in stages 2 and 3, in which it correlates with a more severe disease and a better response to treatment with infliximab. Serum angiotensin converting enzyme (SACE), actively produced by macrophages and involved in granuloma formation, has a low sensitivity and specificity and is not correlated to disease progression in APS [17,56,57]. In contrast, it has been shown, as have serum IL-2 receptors (sIL2Rs), to be correlated to functional improvement under treatment [58]. High levels of sIL2Rs, in contrast, seem to indicate disease progression [59,60]. Chitotriosidase, secreted by macrophages and neutrophils, has markedly increased levels in patients with fibrotic pulmonary sarcoidosis compared to sarcoid patients without [61]. Interestingly, its levels are also significantly correlated with disease outcomes [62]. Finally, it also correlates with FDG-PET/CT findings in Scadding stages 1 to 3 [63]. KL-6, produced by type II epithelial cells, is also increased in fibrotic sarcoidosis but, being also elevated in other interstitial pulmonary diseases, it does not have any reliable diagnostic value in APS [64]. A marked increase in collagen pathway metabolites have been shown in sarcoid patients with sarcoidosis fibrosis compared to those without [65]. Finally, a very interesting study evaluating plasma proteomic markers of progression in fibrotic ILDs found that 17 of them showed consistent associations across all types of ILD subtypes [66]. However, sarcoidosis was not included in the study, and no conclusion can therefore be drawn about APS from these results.

Obviously, further studies are needed to confirm the promising data of some of the biomarkers cited above and to evaluate their potential availability in clinical practice.

### 2.7. Progressive APS

There is no real consensus over the definition of progressive APS, but recent guidelines focusing on all progressive ILDs, besides idiopathic pulmonary fibrosis (IPF), proposed to consider as such an absolute decline over one year of %FVC > 5% pred or of %DLCO > 10% pred, as well as a worsening of imaging and/or clinical symptoms [67]. There is no consensus either over any reliable single marker of disease progression in APS, although numerous studies have aimed at defining how to predict the event in order to ensure the best monitoring and clinical management. Among them, one has compared a large cohort of APS patients with a matched general population during a follow-up of 10 years. It showed that 16 subjects in the first group died during the FU, mainly due to SAPH and respiratory failure [12]. Another one [10] focusing on predictive markers of mortality in sarcoid patients showed that 90% of death cases were related with respiratory failure, with age, extent of fibrosis ≥ 20% on HRCT and SAPH being independent predictors of it. In this cohort, those with >20% fibrosis on HRCT had a mortality significantly higher than those with <20% fibrosis (18.18% vs. 7.35%, respectively, *p* < 0.02). Another one showed that an extent of fibrosis ≥ 20% was associated with a transplant-free survival of 8 years, compared to 17 years in those a pulmonary fibrosis extent <20% [9]. Lung function decline can add a significant value to HRCT findings. Indeed, if a 12-month decline of %FVC > 5% or %DLCO > 10% was not associated with survival, a 2-year lung function decline did predict mortality in a retrospective study [68]. A %DLCO < 30% pred also predicted a clinical worsening event (lung transplantation, death or an absolute drop in %FVC > 10%) [69]. However, no specific single marker seems to be strong enough to predict APS progression, and different clinical as well as imaging scores have been proposed for this purpose. The most sensitive and validated one so far is a composite one, based on both functional (FVC, DLCO and FEV1, resulting in a composite physiological index (CPI)) and 2 HRCT features (extent of fibrosis and diameter of the main pulmonary artery over that of the ascending aorta) [70]. A CPI ≥ 40 was the strongest predictor of survival. Jeny et al. [9] showed in their study that survival in fibrotic pulmonary sarcoidosis was best predicted by geographic origin, lung fibrosis extent, SAPH on echocardiography, or MPAD/body surface area (BSA), as well as by the modified Walsh algorithm. Importantly, a multicenter prospective study in ongoing in biopsy proven pulmonary sarcoidosis to identify markers of progression and the optimal interval for serial clinical monitoring when using the risk prediction markers [71].

In any case, it remains crucial in front of a worsening sarcoid patient and before incriminating a progressive APS to rule out symptoms of potential associated conditions such as cardiac involvement, infection, steroids side effects (obesity, diabetes, myopathy), and psychologically increased symptoms (dyspnea, anxiety, fatigue) [72].

### 2.8. Complications

(a)Bronchiectasis

Present in a large proportion of patients with APS, they can mainly affect central airways through fibrosis related distortion and are prone to become infected partly due to a decreased mucocilliary clearance. All types of immunosuppressants expose patients to infection, particularly the anti-TNF alpha ones [73,74,75].

(b)Infections

Being in themselves a marker of bad prognosis, they are mainly so due to both treatment related immunosuppression and the presence of bronchiectasis. Opportunistic infections can develop in APS, contributing to increased morbidity and mortality [2,76]. The main infectious agents in a large series [77] were Cryptococcus (59%), followed by mycobacteria (13%), nocardia (11%) histoplasma and pneumocystis (9%) and aspergillus (7%) [38]. The most frequent presentation of the latter is chronic cavitary pulmonary aspergillosis [6,78], which can favor the development of a mycetoma presenting as a fungal ball inside a cavitary lesion. Often asymptomatic, but sometimes causing nonspecific signs such as cough and/or dyspnea or even severe signs of infection, it can be life-threatening through massive hemoptysis. Aspergillosis diagnosis is based upon imaging and serological tests, rather sensitive and specific. The presence and levels of galactomannan in bronchoalveolar lavage, sputum or bronchial aspirate is by far less sensitive [79].

(c)Acute exacerbations (AE)

Defined as a clinical and/or functional (>10% decline in FVC and/or FEV1) [80] progressive worsening over a month, they need to be distinguished from other acute complications such as infections and pulmonary embolism, the latter being more frequent in sarcoidosis and affecting particularly more severe and advanced ones [81]. Their true incidence is not known, but a large study of APS reported a frequency of 70% of >2 episodes of AE in the previous year [73]. The majority of patients with APS indeed experience at least one or several AE episodes, if not due to infection or pulmonary embolism then due to bronchospasms, particularly in those with bronchial obstruction and hyperreactivity [73,80]. The use of anti-TNF alpha drugs and the presence of bronchiectasis have been shown to be strong predictors of the risk of AE [73].

(d)Sarcoidosis-associated pulmonary hypertension (SAPH)

Sarcoidosis-associated pulmonary hypertension (SAPH) is a rare but severe complication, mainly occurring in patients with more advanced lung disease. A recent experts statement on its diagnosis and treatment is a significant improvement for its management [82]. Precise SAPH prevalence is difficult to evaluate considering the large variability of populations studied and of the diagnostic methods and criteria used in studies focusing on this topic. If it has been reported with a range as wide as 2 to 74% [83], its overall prevalence may be estimated to 2–6% in unselected sarcoidosis patients, and should be reevaluated in the future, considering that all studies to our knowledge have been produced prior to the newest hemodynamic definition of pulmonary hypertension using a lower mean pulmonary arterial pressure (20 instead of 25 mmHg) [84]. SAPH pathogeny is usually related to multiple and potentially entangled mechanisms. Considering this multifactoriality, the ESC/ERS Task Force and 6th World Symposium on Pulmonary Hypertension have placed SAPH in the WHO group 5 [85]. Indeed, if usually predominantly related to the underlying parenchymal lung disease, SAPH may also involve mechanisms belonging to groups 1, 2, 3 and/or 4. Clinicians should always evaluate the predominant pathogenic mechanism for the sake of the most appropriate management of patients with SAPH. However, patients with APS (particularly fibrotic APS) are more likely to develop a group 3 PH, the most frequent phenotype found in in a recent cohort of 40 subjects [86]. The algorithm leading to SAPH diagnosis includes a full clinical, biological, and functional screening completed by transthoracic echocardiogram (TTE) which allows the determination of SAPH probability, which will be ultimately confirmed by right heart catheterization (RHC). Typical clinical situations requiring SAPH screening include persistent or worsening of dyspnea, especially if there are no signs of parenchymal progression. PFT in case of SAPH usually show a marked alteration of DLco, whether or not associated with a decrease in forced vital capacity (FVC) [87,88], or an isolated DLCO decrease over time [89]. 6MWT also provides important information regarding screening and prognostication of SAPH. Median walk distance is generally <350 m in that case, with a desaturation > 5% [10]. TTE is the best noninvasive diagnostic tool for SAPH through the measurement of the maximum tricuspid regurgitant jet velocity (TRV), now known to be the most relevant measurement for PH estimation. The 2022 ESC/ERS PH guidelines have proposed a simple algorithm that includes TVR and other indirect measurements leading to a score of high, intermediate or low probability for PH [85]. Altogether, although not systematic in all sarcoid patients, TTE should be performed in those with either clinical symptoms, circulating biomarkers (mainly elevated BNP or NT-proBNP), suggestive of PFT or ECG features [82]. RHC remains the only definite diagnostic PH tool. Usually performed in high-risk patients (e.g., with high PH probability evaluated by TTE or in intermediate TTE probability with a low parenchymal involvement as defined by FVC > 59%), it should be discussed on a case-by-case approach if there is a reasonable PH suspicion (e.g., symptoms, PFT, 6MWT, biology and/or imaging). It is mandatory in all patients considered for lung transplantation.

### 2.9. Management

Not standardized yet, it is based upon expert opinion and multidisciplinary discussions aiming at precisely evaluating whether the present case has a relative reversible potential, i.e., an inflammatory component into the pulmonary fibrosis. As stated above, no single functional, imaging or biological marker has a definite value for this diagnosis. The combination of HRCT and PET/HRCT has been so far the best tool for an accurate assessment of this situation [4,90]. In addition, the management of APS should be holistic, addressing all components of patients’ needs in terms of comorbidities treatment and supportive care.

(1)Anti-inflammatory drugs

Their role in APS relies upon the pathogenic inflammation basis of fibrosis in sarcoidosis. They are proposed to treat active inflammation and to prevent the extension of fibrosis. Evidence-based guidelines for treatment of sarcoidosis [4] have proposed the use of steroids as first step agents, combined or not with a steroid-sparing agent, to control APS symptoms. Dose reduction must be started early with the lowest effective doses to avoid side effects and preserve quality of life. Methotrexate is most widely used as the CT sparing drug, or as the second line agent, with the third line one being infliximab, which is superior to all other TNFa inhibitors [91]. Treatment withdrawal after 6–12 months has been associated with a roughly 50% relapse in APS [92]. However, most patients will eventually be put off treatment, mostly because of drug reactions or infections [93]. All other immunosuppressive drugs (azathioprine, mycophenolate mofetil, leflunomide, adalimumab, rituximab, repository corticoids) will be discussed on a case-by-case basis.

(2)Antifibrotic drugs

There is no consensus over the use of these drugs in the context of advanced, fibrotic sarcoidosis, nor over their best timing compared to the anti-inflammatory ones (prior to them? concurrently? after them?). This should be discussed by experts in a MDD on a case-by-case basis.

Nintedanib has been shown in the INBUILD trial to significantly reduce the FVC decline in progressive fibrotic ILD [94]. However, this trial was largely underpowered for sarcoidosis (*n* = 12 among a total of 663 patients), and no prospective trial has been conducted so far on the use of nintedanib in APS, so far leaving the protective effect of nintedanib on fibrotic APS inconclusive. With sarcoidosis having been excluded from a trial testing pirfenidone in PPF [95], there are presently no data to definitely argue the use of this drug in APS. It is presently being tested in an ongoing prospective clinical study on dangerous pulmonary sarcoidosis, i.e., with >20% fibrosis on HRCT [69].

(3)Roflumilast

In patients with repeated episodes of cough and sputum expectoration, a treatment of at least 3 months of this drug led to an improvement of quality of life (QoL) and to less follow-up visits with an FEV1 of less than 90% of best prior value [96].

(4)Treatment of bronchiectasis and infection

General measures are similar to those for other causes of bronchiectasis: mucociliary clearance techniques, sputum cultures surveillance, early antibiotics if signs of infection, and consideration of decreasing immunosuppressants to limit the risk of recurrent infection. There are no specific recommendations for chronic pulmonary aspergillosis. Different antifungal drugs can be used, either systemically (itraconazole, voriconazole, caspofungin) or through percutaneous intracavitary instillations of amphotericin B, with inconsistent results for the latter [97]. Interventional radiologic procedures, such as bronchial artery embolization, might be used in case of massive hemoptysis [98]. Surgical resection might also be indicated and curative, although it exposes to severe per- or post-operative complications due to the poor underlying condition.

(5)Treatment of SAPH

Several potential treatments for SAPH are available, but the optimal management is not defined yet, and the approach to therapy depends on the dominant pathophysiological phenotype, the severity of PH and the severity of the underlying parenchymal lung disease. Because of the lack of robust data, the recent WASOG statement on SAPH [82] has no specific recommendation for pre-capillary treatment, but suggests that an off-label use of PAH therapy may be considered for symptomatic patients on a case-by-case basis after evaluation by a multidisciplinary team with a sarcoidosis and a PH experts. In case of parenchymal involvement, treatment of the inflammatory part of SAPH is supported by limited data but should be considered. However, since the majority of patients presents with a fibrotic disease, immunosuppressive therapy is not likely to be effective alone, and this might argue for the use of specific PH therapies with pulmonary vasodilators. Studies evaluating pulmonary vasodilators in SAPH are small and mostly retrospective. Phosphodiesterase type 5 inhibitors in monotherapy (PDE5i), sildenafil or tadalafil, have been associated in this indication, with an improvement in hemodynamic parameters but not with 6MWT distance [99,100]. Guanylate cyclase stimulators have recently been evaluated in a small prospective study in SAPH, and riociguat has demonstrated a significant improvement in the time to clinical worsening and the 6MWT distance [101]. Endothelin receptor antagonists (ERA) have also been studied in that indication, bosentan prospectively showing improvement in hemodynamics [102] and ambrisentan in the clinical WHO class at 24 months [103], both without modification in the 6MWT. Concerning prostanoid therapies, available studies are also small and retrospective in SAPH [104,105]. Significant improvement in hemodynamics was observed with I.V. epoprostenol in these two studies. The oral selective prostacyclin receptor antagonist selexipag and the promising inhaled treprostinil [106] are currently under evaluation in prospective controlled trials: the SPHINX and the SAPPHIRE trials (ClinicalTrials.gov identifier NCT03942211 and NCT03814317), respectively. The optimal management of SAPH is still pending, and if the use of combination therapy is recommended in group 1 PH, two recent registries of SAPH demonstrate that monotherapies are more frequently prescribed in this indication [107,108]. In all cases, these vasodilators should be used with caution regarding the risk for worsening hypoxemia due to ventilation/perfusion mismatch, and should be avoided in case of pulmonary venous disease or pulmonary veino-occlusive disease. Moreover, patients with an advanced refractory disease should be selected for referral and consideration for lung transplantation (LTx).

(6)Lung transplantation (LT)

A multidisciplinary approach is indeed crucial to the optimal timing for this referral for the experts to anticipate on the comprehensive management of perioperative and post-transplant care [109]. In a study of 1034 candidates for LT listed after the Lung Allocation Score (LSA), 110 had died on the waiting list [110]. Significant predictors of mortality on the waitlist were female gender and severe SAPH, suggesting that these features should prompt clinicians to refer these patients for an early evaluation pf LT possibilities. Survival after LT in sarcoidosis is similar to that of other ILDs, but two factors are predictive of worse survival: older age and the extent of preoperative lung fibrosis [111], suggesting again to take them into account when considering referral to a LT expert center. Interestingly, the LAS has been recently challenged [112]. Indeed, sarcoidosis is grouped within this system with other end-stage lung diseases according to the mean pulmonary artery pressure value: when <30 mmHg (group A sarcoidosis), patients are grouped with COPD, whereas when >30 mmHg (sarcoidosis group D), they are grouped with IPF. This study showed that group D had the highest mortality on the waitlist, and the authors suggested a revision of sarcoidosis LT candidates grouping in allocations systems [112] to mitigate waitlist mortality disparity.

(7)Holistic management of APS

In all cases, the treatment of APS requires a holistic and comprehensive management of all dimensions of the disease (symptoms, psychological distress, fatigue). Some general measures are common to all patients with severe progressive pulmonary fibrosis [113], and comprise age appropriate vaccinations, oxygen supplementation to relieve dyspnea and pulmonary rehabilitation [114,115]. Some measures are more specific of sarcoidosis, such as monitoring of drug toxicity and the management of fatigue.

## 3. Cardiac Sarcoidosis

Cardiac sarcoidosis (CS) is a challenging and underestimated complication that can affect virtually any part of the heart [116,117]. Most frequently, CS is asymptomatic, with only 5–7% of patients manifesting clinical symptoms, with or without the involvement of other organs. Accordingly, CS is often misdiagnosed, particularly when the heart is the only organ affected. Yet, autopsy studies and advanced cardiac imaging suggest a prevalence of CS as high as 20–30% [118]. Cardiac arrest or sudden cardiac death (SCD) may represent the first manifestation of the disease. In a study conducted in Finland from 1998 to 2015, 62 CS cases were diagnosed post-mortem in a cohort of 351 patients. In 11% of them, SCD was the first and only manifestation of CS [119]. Moreover, previous studies have shown that patients presenting with symptomatic CS have a 10% risk of SCD over 5 years of follow-up [120].

The diagnosis of CS is one of the most difficult for several reasons: the lack of universally agreed upon diagnostic criteria may delay the diagnosis and worsen outcomes; endomyocardial biopsies are highly specific in the presence of noncaseating granulomatous inflammation, but the sensitivity of the procedure is as low as 20–25% [121]. According to the Heart Rhythm Society (HRS) statement, CS can be diagnosed histologically from myocardial tissue (definite CS) or clinically (probable CS) in the presence of biopsy proven extracardiac sarcoidosis and of one or more of the following: (i) prompt response to corticosteroids or immunosuppressants, (ii) imaging pattern consistent with CS, (iii) unexplained reduced left ventricular (LV) ejection fraction, and (iv) unexplained major arrhythmic events. In any case, the exclusion of alternative diagnoses remains crucial [122]. The recommendations for diagnosing CS based on the Japanese Circulation Society guidelines are summarized in Table 1 [123].

This makes the diagnosis of isolated CS challenging in cases where endomyocardial biopsy is not available or negative. However, in general, “probable involvement” is considered adequate to establish a clinical diagnosis of CS [122,124,125].

Complete atrioventricular block and ventricular arrhythmias, which may occur in the presence of normal or slightly reduced LV ejection fraction, are rare but life-threatening complications of CS. However, ECG findings, such as atrial arrhythmias, pathological Q waves, and repolarization abnormalities, can be present in up to 90% of cases [126]. For patients younger than 60 years with unexpected high-degree AV block, the guidelines recommend radiological screening for sarcoidosis with cardiac MR or FDG-PET [122]. Heart blocks are the most common presentation of CS due to the frequent localization of granulomas in the basal septum and the involvement of the nodal artery [117]; heart blocks may also represent the cardiac abnormality most likely to respond to steroids. However, in cases of advanced or complete A-V block, pacemaker and cardioverter implantation is highly recommended. As with extracardiac disease, corticosteroids are the mainstay of treatment with escalation to immunosuppressive therapy is when corticosteroid therapy is not tolerated or unable to control the disease. Methotrexate (MTX) is the most widely used second-line agent, with azathioprine and mycophenolate mofetil representing alternative therapeutic choices. Infliximab, a IgG1k monoclonal antibody against TNF-α, may be used for refractory CS [4,127,128,129] (Table 1 and Table 2).

In cases of right ventricular involvement, cardiac sarcoidosis can mimic arrhythmogenic right ventricular cardiomyopathy. Malignant arrhythmias are usually connected to re-entrant pathways caused by myocardial scaring, but triggered activity or dangerous automaticity may also be detected. If extensive, chronic granulomatous inflammation and abnormal remodeling may induce irreversible structural changes leading to progressive cardiomyopathy and heart failure, which manifests as peripheral edema, dyspnea, pulmonary hypertension, and death. Heart failure, whether with low or preserved LV ejection fraction accounts for approximately 25% of deaths in patients with CS.

Cardiac sarcoidosis has an unpredictable course. Early reports indicated a high mortality risk, with 5-year survival of approximately 60% [130,131], whereas more recent series suggest that mortality rates are declining [132], probably because of earlier diagnosis and more efficacious treatments. In a retrospective observational study of 50 patients with CS over a 20-year period, the 1-, 5-, and 10-year event-free survival rates were 96%, 79%, and 58%, respectively [133].

Recent outcome data even indicate 90% to 96% 5-year survival in manifest CS, with the 10-year figures ranging from 80% to 90% [127]. In patients with end-stage cardiac disease, heart transplantation represents a viable option, with patients with CS experiencing post orthotopic heart transplantation survival, odds of graft failure, hospitalization for infection, and post-transplant malignancies similar to patients with non-CS [134].

Overall, patients with CS have reduced health-related quality of life and worse clinical outcomes than patients with extracardiac disease. Moreover, CS patients represent a higher economic burden than those without cardiac involvement [135,136].

## 4. Hepatic Sarcoidosis

Hepatic involvement is a frequent, yet often under-recognized, complication of sarcoidosis; however, a specific treatment is unnecessary in most cases. Autopsy studies found a prevalence of liver sarcoidosis as high as 70% [137,138], while 10–30% of patients display elevated liver enzymes (alkaline phosphatase, gamma glutamyl transferase or serum transaminases) as the only hepatic manifestation of the disease. On abdominal examination, 20% of patients have palpable hepatomegaly, which is frequently associated with splenomegaly [139]. Most patients with hepatic sarcoidosis are asymptomatic, whereas subjects with symptomatic disease (15%) may report diffuse pruritus, fatigue, and right upper quadrant abdominal pain. Around 5% of patients can also present jaundice, weight loss, and fever [140]. A small minority of patients with liver sarcoidosis may develop cirrhosis and end-stage disease, thus needing liver transplantation. Portal hypertension (PH) is reported in about 3–10% of cases, mainly secondary to the compression of portal venules by granulomatous inflammation. Variceal bleeding is a rare but life-threatening complication that requires prompt intervention to avoid hemorrhagic shock and death [141,142,143]. Hepatic encephalopathy is a late and rare complication of liver sarcoidosis, and is typically associated with end-stage disease [144]. In a recent report of 12 cases with symptomatic PH, six patients had esophageal varices complicated by bleeding, whereas only one developed hepatic encephalopathy [145]. Two patients required liver transplantation. In the USA, less than 0.01% of the total number of liver transplants are performed in patients with end-stage liver sarcoidosis [146]; notably, patients transplanted for sarcoidosis tend to have a worse prognosis compared with patients transplanted for indications such as primary biliary cirrhosis or primary sclerosing cholangitis.

Several case reports suggest an association between liver sarcoidosis and risk of cancer, especially hepatocellular carcinoma [147], while lymph node sarcoid reactions have been observed in patients with cholangiocarcinoma [148]. Ursodeoxycholic acid and oral corticosteroids represent the first-line treatment, whereas refractory cases require second-line agents such as methotrexate and azathioprine [149,150]. No randomized controlled trial has evaluated the treatment of hepatic sarcoidosis, and unsolved questions remain regarding who needs treatment and when [145].

## 5. Hypercalcemia and Renal Disease in Sarcoidosis

Renal disease is a rare but potentially life-threatening manifestation of sarcoidosis; it is generally associated with alterations of calcium metabolism or, less frequently, with direct involvement of kidneys or genitourinary tract by granulomatous inflammation. Data about disease prevalence is inconsistent but approximately 1% of all sarcoidosis patients can manifest with overt renal involvement [151,152]. Conversely, if patients are systematically screened, renal disease has been described in up to 25–30% of them, suggesting that most patients have an asymptomatic involvement [153,154].

### 5.1. Alterations of Calcium Metabolism and Its Clinical Manifestations

Hypercalcemia and hypercalciuria are frequent alterations of calcium metabolism, being present in more than half of sarcoidosis patients. In a study of 152 patients, Lebacq and colleagues observed hypercalcemia and excessive urinary calcium excretion in 11% and 62% of them, respectively [155]. The main pathogenetic mechanism is the abnormal production of 1,25-dihydroxyvitamin D3 by macrophages, leading to bone resorption and intestine calcium absorption. Alveolar macrophages are trapped into the alveoli and sustain the granulomatous inflammatory process. This mechanism favors the unbalanced release of the active form of vitamin D [156,157]. Parathyroid hormone (PTH) induced-suppression by increased 1,25-dihydroxyvitamin D3 levels and hypercalcemia lead to hypercalciuria, which, if untreated, can cause renal stones, nephrocalcinosis, and renal failure. Vitamin D therapy, excessive sun exposure and concomitant malignancy may increase the risk of renal sarcoidosis [158]. In a study of 110 sarcoidosis patients, 3.6% of them had nephrolithiasis as the first clinical manifestation, with 2.7% of subject being asymptomatic [159]. Patients with hypercalciuria may have an active, reversible interstitial inflammation and has been suggested that hypercalciuria may predict long-term recovery and better response to immunosuppressive therapy [160].

Nephrocalcinosis is another complication of untreated (or recalcitrant) hypercalcemia, and is characterized by parenchymal calcifications [161]. The diagnosis is generally made by computed tomography (CT), which is able to identify hyperdense tissue with high sensitivity and specificity (92% and 89%, respectively). CT can also reveal the presence of calcium deposits in the renal parenchyma by showing solitary or multiple hyperdense nodules [162].

### 5.2. Glomerular Disease and Interstitial Nephritis from Granulomas

Several types of glomerular lesions have been described in sarcoidosis, although most of the data derives from small case series. An analysis of 26 patients with biopsy-proven glomerular lesions revealed that 42% of them developed glomerulopathy about 10 years after the diagnosis of sarcoidosis, while in 23% and 35% of cases, respectively, glomerular disease developed before and simultaneously with the diagnosis of sarcoidosis. Membranous nephropathy was the most common finding (11/26), followed by IgA nephropathy (6/26), focal segmental glomerulosclerosis (4/26), and, less frequently, nephrotic syndrome and proliferative lupus nephritis (3/26 and 2/26, respectively). Of note, patients with sarcoidosis and concurrent glomerulopathy exhibited most often granulomatous interstitial nephritis (GIN), with poor response to corticosteroid treatment and high risk of developing end-stage renal disease [163].

A study of 27 histologically proven cases of renal sarcoidosis revealed the presence of non-GIN and GIN in 44% and 30% of patients, respectively, and that GIN was more often associated with end-stage renal disease than any other histological pattern of glomerulopathy [164].

### 5.3. Renal Failure

Renal failure is an uncommon manifestation of sarcoidosis, and can result from arteriolar nephrosclerosis, hypercalcemia, nephrocalcinosis, interstitial nephritis and glomerulonephritis; conversely, renal granulomatous inflammation, though present in up to 40% of autopsy studies, is an uncommon cause of renal failure [165]. Vitamin D replacement and malignancy can impair further renal function, leading to hypercalcemia and nephrocalcinosis [166,167]. Renal failure as the first manifestation of sarcoidosis is less frequently observed. Pastor and co-workers reported on a case of early-onset renal failure, with the patient family tree revealing a strong history of sarcoidosis with two confirmed and two possible cases. An overlap with other autoimmune disorders, such as ulcerative colitis, Graves’ disease and coeliac disease, was found in these patients [166].

## 6. Conclusions

High-risk sarcoidosis features are now better defined, but still lack well established predictive markers. Strongly impacting mortality and morbidity, it should be diagnosed early and if possible prevented by appropriate therapeutic measures. The latter are, however, not definitely established. The role of anti-inflammatory drugs and of antifibrotic ones remain to be confirmed through clinical trials. That of the requirement of a holistic management of these patients is in contrast absolutely consensual. The need for powerful preventive treatments of such high-risk situations paves the way for hopefully upcoming and closely evaluated new approaches such as “Hit-hard and early versus step-up treatment in severe sarcoidosis” [168] and/or for new drugs (Efzofitimod and JAK inhibitors, for instance).

## Figures and Tables

**Table 1 diagnostics-14-00395-t001:** Japanese Circulation Society 2017 Criteria for the Diagnosis of Cardiac Sarcoidosis [122].

Histological diagnosis: Endomyocardial biopsy or surgical specimens demonstrate non-caseating epithelioid granulomas and other causes and local sarcoid reactions can be ruled out
Clinical diagnosis (1) Granulomas are found in organs other than the heart, and clinical findings are strongly suggestive of cardiac involvement, or (2) clinical findings are strongly suggestive of pulmonary or ophthalmic sarcoidosis and at least 2 of the 5 characteristic laboratory findings of sarcoidosis are present and clinical findings strongly suggest CS: Two or more of the five major clinical criteria are satisfied, or One of five of the major criteria and two or more of the three minor criteria are satisfied Major criteria: (1) High-grade atrioventricular block (including complete atrioventricular block) or fatal ventricular arrhythmia; (2) Basal thinning of the ventricular septum or abnormal ventricular wall anatomy (ventricular aneurysm, thinning of the middle or upper ventricular septum, regional ventricular wall thickening); (3) Left ventricular contractile dysfunction (ejection fraction less than 50%) or focal ventricular wall asynergy; (4) ^67^Ga-citrate scintigraphy or ^18^F-FDG PET reveals abnormally high tracer accumulation in the heart; (5) Gadolinium-enhanced CMR reveals delayed contrast enhancement of the myocardium. Minor criteria: (1) Abnormal ECG findings: non-sustained ventricular arrhythmias or multifocal or frequent premature ventricular contraction, bundle branch block, axis deviation, abnormal Q waves; (2) Perfusion defects on myocardial perfusion scintigraphy; (3) Endomyocardial biopsy: monocyte infiltration and moderate or severe myocardial interstitial fibrosis.
Isolated cardiac sarcoidosis: (1) No clinical findings characteristics of sarcoidosis are observed in any organs other than the heart; (2) ^67^Ga-scintigraphy or ^18^F-FDG PET reveals no abnormal tracer accumulation in any organs other than the heart; (3) A chest CT scan reveals no shadow along the lymphatic tracts in the lungs or no hilar and mediastinal lymphadenopathy (minor axis > 10 mm) and Endomyocardial biopsy or surgical specimens demonstrate non-caseating epithelioid granulomas, or ^67^Ga-scintigraphy or ^18^F-FDG PET reveals abnormally trace accumulation are present in heart and at least three other major criteria are satisfied, or Endomyocardial biopsy or surgical specimens demonstrate non-caseating epithelioid granulomas, or ^67^Ga-scintigraphy or ^18^F-FDG PET reveals abnormal tracer accumulation is present in the heart and at least three other major criteria are satisfied

**Table 2 diagnostics-14-00395-t002:** Features that influence treatment decision making in cardiac sarcoidosis [117].

Age > 50 Years
Left ventricular ejection fraction < 40%
New York Heart Association Functional Class III or IV
Increased left ventricular end-diastolic diameter
Late gadolinium enhancement on cardiac MRI
Ventricular tachycardia
Cardiac inflammation identified by FDG-PET
Echocardiographic evidence of abnormal global longitudinal strain
Interventricular septal thinning
Elevated troponin or brain natriuretic peptide

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
