# Peer review of "High-Risk Sarcoidosis: A Focus on Pulmonary, Cardiac, Hepatic and Renal Advanced Diseases, as Well as on Calcium Metabolism Abnormalities"

_diagnostics, 2024, doi:10.3390/diagnostics14040395_

Round 1

Reviewer 1 Report

Comments and Suggestions for Authors

The manuscript addresses very important aspects of sarcoidosis, the ones that generate the most difficult to manage situations related to sarcoidosis, and that have insufficient solutions. 

The manuscript is well structured and includes the best of current knowledge but also authors' ideas about the future development of the topic.

I consider the manuscript very useful for practitioners dealing with severe sarcoidosis cases.

Author Response

Answers to reviewers

We are very grateful to reviewer 1 for its reading of our manuscript «High-risk sarcoidosis : a focus on pulmonary, cardiac, hepatic and renal advances disease as well as on calcium metabolism abnormalities » by Dominique Israël-biet et al.

Reviewer 2 Report

Comments and Suggestions for Authors

In this review, they authors discuss issues pertaining to diagnosis and treatment of advance/chronic sarcoidosis, focusing on progressive pulmonary disease, heart, liver, and kidney disease including hypercalcemia and hypercalciuria. The review is of interest to physicians and researchers, is quite informative and, except for a few minor issues, well-written.

There are, however, some minor issues that the authors should address to improve their manuscript. I list them as they appear in the main text:

1.     Line 24: “Darker” should be replaced by “worse”.

2.     Line 63–64: Why is only a mixed defect associated with reduced DLCO, and stage 4 sarcoidosis? Would not the same apply to restrictive disease alone. The authors should clarify. Also, sarcoidosis staging should be defined to avoid any confusion. I guess they refer to Scadding stages.

3.     Line 77: Stage 3 and 4 should per definition not involve hilar lymph nodes, or do the authors refer to isolated parenchymal disease? Kindly, clarify.

4.     Line 203: What was the proportion of patients with signs/diagnosis of some kind of aspergillosis in this study?

5.     Line 228–229: This/these sentence(s) should be reviewed here. Something seems to be missing.

6.     Line 236: “fibrotic one” should read “fibrotic APS” so it is not confused with ‘a patient with fibrosis’.

7.     Line 272: “associated with a steroids sparing one”. I guess the authors mean ‘combined or not with a steroid-sparing agent’.

8.     Line 385–386: The authors should clarify what constitutes/differentiates ‘definite’ and ‘probable’ cardiac sarcoidosis diagnosis according to the HRS criteria. Also, do the authors have any opinion if HRS or the Japanese criteria are better for diagnosing cardiac involvement?

9.     Line 417: “poor prognosis” comes into contrast with the survival probabilities provided in the next sentence. There is no reference/citation to support this statement. What constitutes poor prognosis and why did this change according to the more recent data that the authors provide?

Comments on the Quality of English Language

Please refer to the main comments.
